# Respiratory Parameters for the Classification of Dysfunctional Insulin Secretion by Pancreatic Islets

**DOI:** 10.3390/metabo11060405

**Published:** 2021-06-21

**Authors:** Uma D. Kabra, Charles Affourtit, Martin Jastroch

**Affiliations:** 1Division of Metabolic Diseases, Technische Universität München, 80333 Munich, Germany; uma.kabra16205@paruluniversity.ac.in; 2Department of Pharmaceutical Chemistry, Parul Institute of Pharmacy, Parul University, Vadodara 391760, India; 3Faculty of Health, School of Biomedical Sciences, University of Plymouth, Plymouth Science Park, Plymouth PL6 8BU, UK; charles.affourtit@plymouth.ac.uk; 4Department of Molecular Biosciences, The Wenner-Gren Institute, The Arrhenius Laboratories F3, Stockholm University, SE-106 91 Stockholm, Sweden; 5Helmholtz Diabetes Center, Institute for Diabetes and Obesity, Helmholtz Zentrum München, German Research Center for Environmental Health (GmbH), 85764 Neuherberg, Germany

**Keywords:** mitochondria, bioenergetics, glucose-stimulated insulin secretion, pancreatic islets, respiration

## Abstract

The development of obesity and type 2 diabetes (T2D) has been associated with impaired mitochondrial function. In pancreatic beta (β) cells, mitochondrial energy metabolism plays a central role in triggering and controlling glucose-stimulated insulin secretion (GSIS). Here, we have explored whether mitochondrial bioenergetic parameters assessed with Seahorse extracellular flux technology can quantitatively predict insulin secretion. We metabolically stressed male C57BL/6 mice by high-fat feeding (HFD) and measured the glucose sensitivity of islet respiration and insulin secretion. The diet-induced obese (DIO) mice developed hyperinsulinemia, but no pathological secretory differences were apparent between isolated DIO and chow islets. Real-time extracellular flux analysis, however, revealed a lower respiratory sensitivity to glucose in DIO islets. Correlation of insulin secretion with respiratory parameters uncovers compromised insulin secretion in DIO islets by oxidative power. Normalization to increased insulin contents during DIO improves the quantitative relation between GSIS and respiration, allowing to classify dysfunctional properties of pancreatic insulin secretion, and thereby serving as valuable biomarker for pancreatic islet glucose responsiveness and health.

## 1. Introduction

Pancreatic β cells are specialized endocrine cells that integrate signals from glucose and other fuels to control the secretion of insulin [1]. Glucose induces insulin secretion via both triggering and amplifying pathways [2]. The triggering pathway involves oxidative glucose catabolism, a rise in the cytosolic adenosine triphosphate/adenosine diphosphate (ATP/ADP) ratio, closure of ATP-sensitive potassium (K_ATP_) channels, depolarization of the plasma membrane, influx of calcium ions, and eventual exocytosis of insulin-containing granules [3,4,5]. The amplifying pathways boost this glucose-stimulated insulin secretion (GSIS) in a K_ATP_-channel-independent (but Ca^2+^-dependent) way [4,6]. Many steps of the β cell insulin secretory pathways can affect the efficiency of insulin secretion. Mitochondria are intimately involved in glucose catabolism because oxidative phosphorylation has control over the ATP/ADP ratio and, thus, GSIS [7]. Oxidative phosphorylation is ideally assessed by combining respiratory flux and mitochondrial membrane potential measurements [8,9]. While mitochondrial membrane potential measurements of β cell mitochondria are challenging in intact β cells because of glucose-evoked plasma membrane potential fluctuations [10], cellular respiratory flux measurements are relatively straightforward [11]. Several parameters are readily calculated from whole-cell oxygen uptake to give insight in the efficiency by which energy liberated from oxidative glucose breakdown is conserved as ATP. These parameters include glucose sensitivity (GS), i.e., the magnitude of respirational increase upon glucose stimulus, and coupling efficiency (CE) of oxidative phosphorylation, i.e., the part of mitochondrial respiration coupled to ATP synthesis [8]. Notably, these bioenergetic parameters are internally normalized and, thus, dimensionless, which renders them more sensitive indicators of oxidative phosphorylation than absolute oxygen uptake rates [8]. For example, CE has been instrumental for demonstrating a glucose sensitivity of oxidative phosphorylation in INS-1E insulinoma cells that is regulated by mitochondrial uncoupling protein-2 [12]. This bioenergetic regulation is reflected by insulin secretion activity, which suggests the possibility that CE has broader predictive power that may be exploited to forecast β cell dysfunction.

In this study, we correlated mitochondrial respiration and insulin secretion in pancreatic islets from chow-fed and diet-induced obese (DIO) mice, with the aim to identify parameters that allow classification of insulin secretion deficiencies. DIO mice represent a model of a well-established risk scenario in which obesity leads to insulin resistance, β cell dysfunction, and eventually type 2 diabetes mellitus [13]. Our findings suggest that loss of mitochondrial respiratory sensitivity to glucose is an early warning sign for compromised insulin secretion. 

## 2. Results and Discussion

### 2.1. Inducing Metabolic Stress in Mice

We tested our hypothesis on the relation between respiration and insulin secretion in DIO mice, a typical model for the development of insulin resistance and hyperinsulinemia. Male C57BL/6 mice were kept on high-fat diet (HFD) for 16 weeks to induce obesity (~45.8 g HFD vs. ~28.8 g chow; Figure 1A). Plasma insulin levels were dramatically increased (Figure 1B). However, plasma glucose levels were unchanged (Figure 1C), suggesting that hyperinsulinemia was sufficient to compensate insulin resistance in HFD mice.

### 2.2. Respiratory Activity but Not Insulin Secretion Is Impaired in Islets of DIO Mice

Interestingly, the assessment of insulin secretion between chow and DIO islets revealed no differences (Figure 2A). In contrast, averaged real-time oxygen consumption traces in Figure 2B demonstrate clearly that respiration in DIO islets differs drastically from that exhibited by their control counterparts. After normalizing oxygen uptake to DNA content and correcting respiratory activity for non-mitochondrial (i.e., rotenone-and-antimycin-A-resistant) oxygen consumption, it transpired that there was no difference in basal respiration between the two systems (Figure 2C). However, the strong respiratory stimulation provoked by glucose in control islets was more than halved in DIO islets (Figure 2D,E), which was the combined result of attenuated respiration linked to mitochondrial proton leak (Figure 2F) and ATP synthesis (Figure 2G). In other words, both oligomycin-sensitive and -insensitive oxygen uptake was decreased by DIO, which explains why the CE of oxidative phosphorylation, which reflects the oligomycin sensitivity of overall glucose-stimulated respiration (GSR), was not affected to a statistically significant degree (Figure 2H). The reduction of both proton leak and ATP-linked respiration can be indicative of compromised substrate oxidation capacity [8]. Next, we plotted absolute secreted insulin values vs respiratory parameters related to insulin triggering (ATP-linked respiration and GSR), and assessed their relation by correlation analysis. Within chow islets, insulin values correlate significantly with ATP-linked respiration and GSR (Figure 2I,J). This relationship appeared to be shifted upwards in DIO islets (dotted regression line). In our experimental conditions in vitro, however, no secretagogues or other amplifying mechanisms are present, to the best of our knowledge, which could selectively act on DIO islets. However, we cannot formally exclude amplifying factors deriving from glucose catabolism [14], and we thus refer to altered secretory pathways of insulin secretion rather than to altered triggering of insulin secretion.

### 2.3. Compensating Increase of Insulin Content in DIO Islets Masks Defects of the Triggering Pathway 

Exploring compensatory responses to loss of insulin sensitivity in HFD mice (Figure 1), we found a 15% higher insulin content in DIO islets (Figure 3A), consistent with increases in β cell proliferation during DIO [15]. Normalizing insulin secretion to insulin content, GSIS of the DIO islets was lower than that from control islets (Figure 3B). Importantly, plotting GSIS values normalized to insulin content shifted DIO insulin values onto the regression of chow islets, with a better regression to glucose-stimulated than to ATP-linked respiration (Figure 3C,D). Insulin secretion correlates less well with coupling efficiency (CE, Figure 3E), but it should be noted that the internal standardization benefits data comparisons between different experimental settings and laboratories. The normalization-dependency of the GSIS phenotype suggests that DIO islets respond to an obesity-induced drop in glucose sensitivity by increasing insulin content, a response that appears to fix the secretory impairment.

### 2.4. Classifying Defects in Insulin Secretion

From Figure 3C–E, it transpires that secreted insulin values require normalization to insulin content to establish a robust linear relationship between insulin and bioenergetics parameters. The relationship between oxidative phosphorylation and GSIS leads to a simple linear correlation model to classify defects of insulin secretion (Figure 3F). In relation to control values, descending along the regression line suggests reduced oxidative power by either compromised substrate delivery or respiratory dysfunction. In the case of DIO islets, the secretory pathway is compromised by reduced oxidative power, which could be mediated by impaired glucose uptake, glycolysis, or pyruvate oxidation—the latter being recently suggested in response to impaired mitochondrial dynamics [16] or inflammation [17]. In contrast, ascending values suggest improved substrate delivery or oxidative phosphorylation. Upwards deviation from regression is explained by amplifying pathways, while downwards deviation points towards secretory dysfunction downstream or no mitochondrial impact.

## 3. Materials and Methods

Animals—male C57BL/6 mice with an age of 8 to 10 weeks were purchased from Janvier Labs (Le Genest-Saint-Isle, France). The animals were maintained on a 12/12 h light/dark cycle in a temperature-controlled environment and allowed free access to standard chow diet (5.6% fat, LM-485, Harlan Teklad) or a high-fat diet (HFD) (58% kcal fat; Research Diets Inc., New Brunswick, NJ, USA) for 16 weeks. All in vivo procedures were conducted under the guidelines of the Institutional Animal Care Committee of the Helmholtz Center Munich, which approved all animal maintenance and experimental procedures. The animal experiments complied with all ethical regulations for animal testing and research, including animal maintenance and experimental procedures that the animal welfare authorities of the local animal ethics committee of the state of Bavaria (Regierung Oberbayern) approved in accordance with European guidelines.

Islet isolation and culture—mouse islets were isolated by digestion with collagenase as described elsewhere [12]. Around 150–200 islets were obtained per mouse. Islets were incubated overnight in RPMI 1640 culture medium supplemented with 10% (*v/v*) fetal calf serum (Life technologies) at 37 °C and 5% CO_2_ before experimentation. 

Insulin secretion—groups of eight size-matched islets were handpicked into individual wells of V-bottomed 96-well plates and incubated for 60 min at 37 °C in HEPES-balanced Krebs-Ringer (KRH) bicarbonate buffer containing 114 mM NaCl, 4.7 mM KCl, 2.5 mM CaCl_2_, 1.16 mM MgSO_4_, 1.2 mM KH_2_PO_4_, 25.5 mM NaHCO_3_, 20 mM HEPES (pH 7.2–7.4), supplemented with 0.2% (*w/v*) bovine serum albumin (BSA), and 2 mM glucose. The islets were then incubated for a further 60 min in KRH-bicarbonate buffer containing 2 or 16.5 mM glucose. Subsequently, supernatants were collected to quantify secreted insulin. Following the secretion assay, islets were lysed with ice-cold RIPA buffer to allow total insulin content to be determined. Insulin was detected with the Ultra-Sensitive Mouse Insulin Elisa Kit (ALPCO, Salem, NH, USA). Data were normalized to DNA content measured with the Quant-it Pico Green DNA assay kit (Invitrogen, Darmstadt, Germany).

Mitochondrial respiration—oxygen consumption rates (OCR) were measured in islet-capture plates of the XF24 extracellular flux analyzer (Agilent, Seahorse Bioscience, Santa Clara, CA, USA). Briefly, groups of 30 size-matched islets were handpicked into individual wells of islet capture plates and incubated for 60 min at 37 °C without CO_2_ in KRH-bicarbonate free buffer containing 2 mM glucose. Additional glucose (16.5 mM), oligomycin (10 µg/mL), and a mixture of rotenone and antimycin A (both 2 µM) were injected sequentially. Mitochondrial respiration was calculated by subtracting non-mitochondrial respiration from all other oxygen uptake rates. The individual bioenergetics parameters of OXPHOS parameters were calculated as follows: basal mitochondrial respiration = (last rate measured before glucose and/or other secretagogues injection) − (non-mitochondrial respiration rate). Glucose-stimulated mitochondrial respiration = (last rate measured before oligomycin injection) − (non-mitochondrial respiration rate). Proton-leak-linked respiration = (minimum rate measured after oligomycin injection) − (non-mitochondrial respiration rate). ATP-synthesis-linked respiration = (glucose-stimulated mitochondrial respiration) − (proton leak-linked respiration). Coupling efficiency = the fraction of respiration used to drive ATP synthesis for each run, calculated as CE = 1 − (proton-leak-linked respiration/mitochondrial glucose-stimulated respiration). Data were normalized to DNA content.

Metabolic tests—mice were fasted for six hours prior to blood collection. Blood samples taken from the tail vein were used to measure glycaemia with a glucometer (Abbott, Wiesbaden, Germany) and insulin by ELISA.

Statistical analysis—statistical analysis was performed with GraphPad Prism version 6.0. The data passed normality tests (Shapiro–Wilk and Kolmogorov–Smirnov test) and the groups were compared using unpaired Student’s *t* test. All the data are shown as mean ± standard error of mean (S.E.M). *p* values < 0.05 were considered statistically significant.

## 4. Conclusions

In this study, we demonstrate that mitochondrial respiratory parameters have predictive value for insulin secretion from pancreatic β cells. In a typical model for metabolic disease, DIO mice, we found that defective insulin secretion is compensated by increased insulin content. Although absolute values of insulin secretion are not affected, mitochondrial respiration is severely compromised in DIO islets. Correlation of respiration and GSIS is firmly established by normalizing GSIS to insulin content, showing that mitochondrial respiratory parameters quantitatively predict changes in GSIS. Absolute values of ATP-linked respiration and GSR predict insulin release the best, while CE is less predictive. CE, however, is a powerful bioenergetic parameter that has been successfully applied to uncover molecular mechanisms in mitochondria, e.g., the role of UCP2 in β cells [8]. CE is based on the thermodynamic laws of energy conversion that mitochondrial oxidation energy is either converted to ATP or lost as heat due to the mitochondrial proton leak. CE is defined as the fraction of energy that is converted to ATP (thus ranging from 0 to 1). As internally standardized parameter, CE is not prone to variation in absolute values between independent experiments. Although CE correlates with GSIS, the relation is steeper and more variable concerning linear regression, as compared to ATP-linked respiration and GSR in our experimental setup. Thus, CE may only be used to compare independent studies. However, the relation of CE and GSIS suggests a “threshold” CE value that is required to trigger insulin secretion. Considering previous studies from our laboratory [16,18], we find a tight CE of about 0.4–0.6 as requirement for insulin triggering in islets and in β cell models. Based on the results we designed a model correlating GSR or ATP linked respiration vs. insulin secretion that allows classifying dysfunctional properties of pancreatic insulin secretion under pathological conditions. All these analyses suggest that mitochondrial bioenergetic parameters reflect insulin secretion in a quantifiable manner, and may thus serve as biomarkers for glucose responsiveness and pancreatic islet health.

## Figures and Tables

**Figure 1 metabolites-11-00405-f001:**
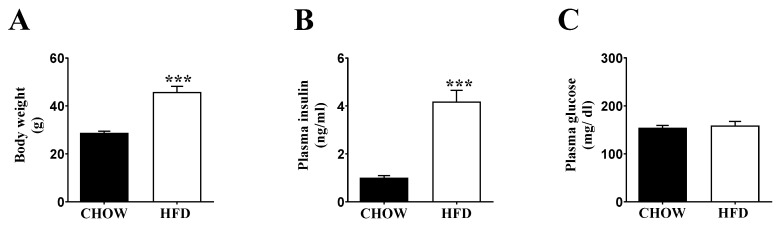
Characterization of chow (black bars) and HFD (white bars) fed C57BL/6 mice. (**A**) Body weight; (**B**) plasma insulin level; and (**C**) plasma glucose levels. The mice were 8 weeks old at the start of the feeding experiment. Data are represented as ± SEM (*n* = 8 mice per group). Statistical significance of mean differences was tested by unpaired two-tailed student *t*-test. *p* < 0.001 (***).

**Figure 2 metabolites-11-00405-f002:**
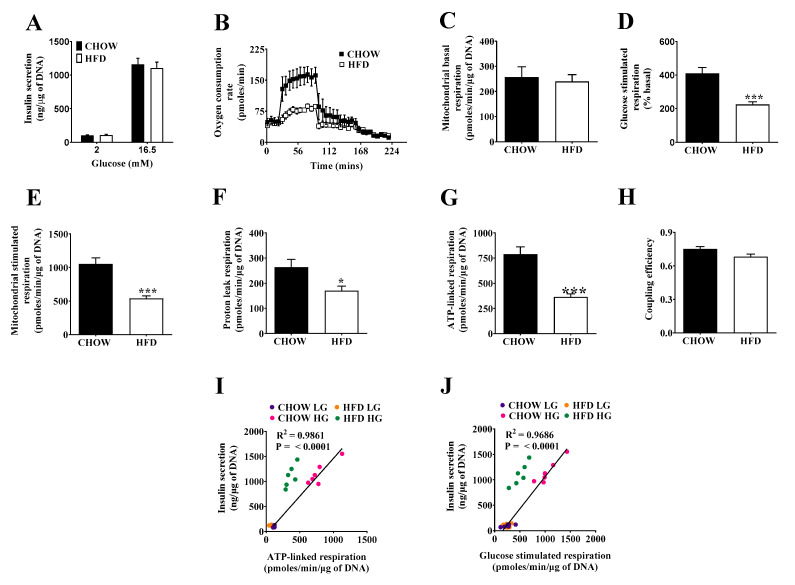
Glucose-stimulated insulin secretion and mitochondrial bioenergetics of pancreatic islets from chow and HFD-fed mice. (**A**) Insulin secretion. Batches of eight size-matched islets were exposed to either 2 or 16.5 mM glucose for 1 h before supernatants and lysates were collected. (**B**) Representative time-resolved oxygen consumption traces. Batches of 30 size-matched islets were exposed to 2 mM glucose to assess basal respiration (4 cycles), glucose-stimulated respiration (10 cycles), proton leak respiration by inhibition of ATP synthase using oligomycin (10 cycles), and non-mitochondrial respiration by final injection of rotenone/antimycin A. (**C**) Mitochondrial basal respiration at low glucose (2 mM). (**D**) Glucose-stimulated respiration expressed as percentage of basal (**E**) Mitochondrial stimulated respiration at high glucose (16.5 mM). (**F**) Proton leak respiration at high glucose. (**G**) ATP-linked respiration at high glucose. (**H**) Coupling efficiency at high glucose. (**I**) Correlation of insulin secretion (absolute values) and ATP-linked respiration. (**J**) Correlation of insulin secretion (absolute values) and glucose-stimulated respiration. Data are represented as means ± SEM (*n* = 6 mice per group, each mouse was considered as independent experiment and islets were plated in triplicate). Statistical significance of mean differences was tested by unpaired two-tailed student *t*-test. *p* < 0.05 (*), *p* < 0.001 (***).

**Figure 3 metabolites-11-00405-f003:**
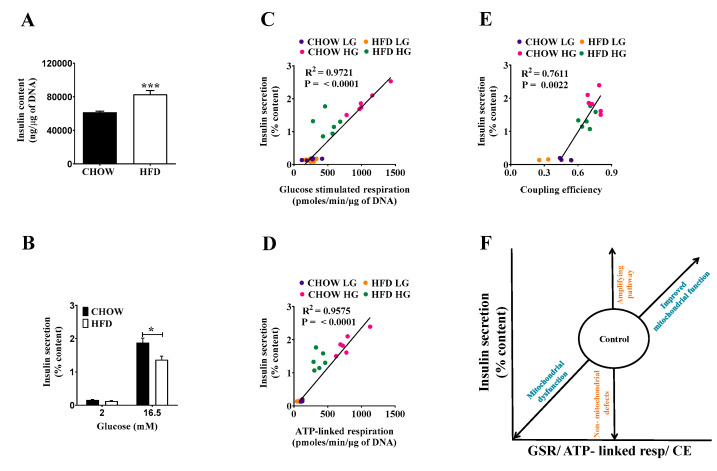
Glucose-stimulated insulin secretion in chow and HFD mice. Batches of eight size-matched islets were exposed to either 2 or 16.5 mM glucose for 1 h at which point supernatants and lysates were collected for (**A**) insulin content, and (**B**) secreted insulin measurements. Correlation between insulin secretion (% content) and (**C**) glucose-stimulated respiration, (**D**) ATP-linked respiration, and (**E**) coupling efficiency (**F**) correlation model classifying defects of insulin secretion. Data are represented as means ± SEM (*n* = 6 mice per group, each mouse was considered as independent experiment and islets were plated in triplicate). Statistical significance of mean differences was tested by unpaired two-tailed student *t*-test. *p* < 0.05 (*), *p* < 0.001 (***).

## Data Availability

Not applicable.

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
