# Peer review of "Respiratory Parameters for the Classification of Dysfunctional Insulin Secretion by Pancreatic Islets"

_metabolites, 2021, doi:10.3390/metabo11060405_

Round 1

Reviewer 1 Report

Dear Authors,

This paper concerned problem of cell respiration, mitochondrial function, and glucose-stimulated insulin secretion in case metabolically stressed mice. This is interesting and potentially important topic. Paper is written using very specialistic language, because of that is difficult to read for normal reader, not introduced to laboratory work.  Additionally, I have a few questions and comments:

  1. In the introduction- In the first using, for not specialistic readers, such terms as glucose sensitivity and coupling efficiency should be explained and define, especially the method of calculation.
  2. Methods- there is lack of number of used mice in the studied and control groups.
  3. Methods- in the islet isolation- How many islets do You menage to isolate from one animal. Did You compare group of 8 islet from HFD and 8 from control, or there were several (how many?) experiments 8 islet in each?
  4. Statistical analysis- There is no information of data normality of distribution and variancy. If it was not done it should be analysed using nonparametric analysis, with correction for small number, not student-t test.
  5. If we do not know the normality of distribution we do not know if showing in mean, and SE are proper to description of data- maybe median, and range will be better.

Author Response

We thank the reviewer for her/his valuable and constructive comments that we addressed point-by-point below.

Reviewer 1

Dear Authors,

This paper concerned problem of cell respiration, mitochondrial function, and glucose-stimulated insulin secretion in case metabolically stressed mice. This is interesting and potentially important topic. Paper is written using very specialistic language, because of that is difficult to read for normal reader, not introduced to laboratory work. 

RESPONSE: We thank the reviewer for his/her thorough reading, positive response and have now amended the specialist language to make this piece better readable also for other scientists in the field of diabetes and islet biology.

Additionally, I have a few questions and comments:

  1. In the introduction- In the first using, for not specialistic readers, such terms as glucose sensitivity and coupling efficiency should be explained and define, especially the method of calculation.

RESPONSE: Reviewer’ point well taken. We have now included a few explaining words on the terms in the introduction (page 2, line 55-57: ‘These parameters include glucose sensitivity (GS) i.e., the magnitude of respirational increase upon glucose stimulus, and coupling efficiency (CE) of oxidative phosphorylation i.e., the part of mitochondrial respiration coupled to ATP synthesis [8].’.

In the methods section, we added the description of calculation for coupling efficiency and other respiratory parameters (page 6 lines 194-203: ‘The individual bioenergetics parameters of OXPHOS were calculated as follows: Basal mitochondrial respiration = (last rate measured before glucose and /or other secretagogues injection) – (non-mitochondrial respiration rate). Mitochondrial respiration = (last rate measured before oligomycin injection) – (non-mitochondrial respiration rate). Proton leak-linked respiration = (minimum rate measured after oligomycin injection) - (non-mitochondrial respiration rate). ATP-linked respiration = mitochondrial respiration – proton leak-linked respiration. Coupling efficiency = represents the fraction of respiration used to drive ATP synthesis for each run, calculated as CE = 1- (proton leak-linked respiration / mitochondrial stimulated respiration). ‘

  1. Methods- there is lack of number of used mice in the studied and control groups.

RESPONSE: We apologize for this short-coming and have now stated the number of mice. The data were collected from eight CHOW and eight HFD mice (Fig. 1 noted in the caption) and for remaining figures, data were collected from six CHOW and six HFD mice (noted in the caption). The added text can be found on page 3 line 82-83 ‘Data are represented as ± SEM (n=8 mice per group).’, on page 4 lines 120-122 ‘Data are represented as ± SEM (n=6 mice per group, each mouse was considered as independent experiment and islets were plated in triplicate) and on page 5 lines 157-159 ‘Data are represented as ± SEM (n=6 mice per group, each mouse was considered as independent experiment and islets were plated in triplicate).

  1. Methods- in the islet isolation- How many islets do You menage to isolate from one animal. Did You compare group of 8 islet from HFD and 8 from control, or there were several (how many?) experiments 8 islet in each?

RESPONSE: From one mouse we were able to get around 150-200 islets (now found on page 6 line 173). Six mice per group were used for the insulin secretion and seahorse experiments. Each mouse was considered as an independent experiment, and islets were plated in triplicates (8 islets for insulin secretion per well and 30 islets per seahorse well, and an average was calculated). We have now included this information in the MS page 4 lines 120-122 and page 5 lines 157-159.

  1. Statistical analysis- There is no information of data normality of distribution and variancy. If it was not done it should be analysed using nonparametric analysis, with correction for small number, not student-t test.

RESPONSE: The reviewer point is well-taken. The data passed normality tests (Shapiro-Wilk and Kolmogorov-Smirnov test) and the groups were compared using unpaired Student’s t test. Now we have included the statement on the tests in the methods, page 6 lines 208-209.

Reviewer 2 Report

In this work, Kabra et al. tested the hypothesis of correlation between respiratory rate and insulin secretion in pancreatic islets to evaluate dysfunctional aspect occurring during T2DM development. In my opinion, this work has important descriptive aspects that has an impact in the community studying metabolic alteration of pancreatic islets during T2DM and obesity development in relation of the ability to secrete insulin upon stimulation. The manuscript is enjoyable to read and well written. The introduction as well as the methods are clear and adequate to introduce the results. The hypothesis is well explained and is addressed very well in the discussion.

Very interesting and novel to my knowledge is the notion that after isolation of islets from DIO mice the GSIS is similar to control diet while the respiration in response to glucose stimulation is still affected. This work shows also that different ways to normalize the collected data might affect importantly the conclusion drawn by the same data. Together with the data showing an increase in insulin content (which might be due as the authors suggest by the increase in beta cells but also an increased insulin translation due to workload), the aspect studied here are often neglected but of high importance when studying the dysfunction of insulin secretion in DIO in mice. Moreover, the author clarify very well which aspect to consider (dimensionless data not needed to be normalized on cell mass or DNA which can increase the data spread) and which one can be predictive of dysfunction in these diet model.

I have few minor points that in my opinion need to be addressed before publication:

  • Please clarify which point you considered for the calculations of proton leak respiration and CE. Is it the maximal respiration after glucose stimulation (single highest point) and the lowest oligomycin point or is an average of several measurements?
  • Lane 28, 35, 53 β cells is spelled using β while in the rest of the text beta cells. I would suggest to be consistent in the nomenclature
  • Lane 68. If the data were collected after the different diets were given for 16 weeks, can you clarify the reason of writing “for a maximum of 16w”.
  • Lane 73. How long those islets were cultured before performing the Seahorse and insulin secretion experiments? This is a very critical point and need to be present in the method. In fact, in our experience after overnight culture the islets from DIO mice regain insulin secretion similar to control (when normalized on DNA) as the authors shows here, but retain some dysfunction if the insulin secretion is performed less than 4h after isolation. This point is important also to understand how long the islets from DIO mice retain dysfunctional aspect on their metabolism or if they can recover by longer exposure in vitro culture.
  • Line 109 Please add if the insulin and glucose were measured in fasting or non-fasting.
  • Lane 113 and 163 please correct normal (diet) with Chow.
  • Could the author speculate why in DIO mice islets also the proton leak respiration is lower than controls?
  • Lane 140 The letter (I) can be placed at the start of the sentence “Correlation…”.
  • Lane 145 the authors suggest that “the insulin values correlates significantly with ATP-linked respiration and GSR”. Wouldn´t be more suitable to say that the correlation between these parameter in chow diet and DIO mice are different? Along this line I would suggest, if appropriate for the authors, to add a dotted line with the regression of the DIO mice islets to show the difference compared to the chow regression in a visual format.
  • Line 175 The authors suggest that “the triggering pathway in DIO mice is compromised by reduced oxidative power, which could be mediated by impaired glucose uptake, glycolysis or pyruvate oxidation…”. I have some considerations here, which can be also included in the results or discussion. 1. The authors did not investigate the impairment of glucose uptake, glycolysis or pyruvate oxidation using an uncoupler (in separate experiment and not after glucose or oligomycin addition) which might have given an answer to this question thus avoiding the ATP/ADP regulation of substrate oxidation/utilization for cellular respiration. In fact, if the DIO mice are set to a higher ratio of cytosolic or mitochondria ATP/ADP ratio the substrate delivery and oxidation might be lower as shown here. 2. The insulin docked and secreted in the first phase is usually measured in the first 10-15min after glucose stimulation. Here the authors measure the insulin secretion after a static incubation of 1h and suggest that the triggering pathway is compromised. However this length of insulin secretion would include also the amplify pathway even though only glucose is present in the assay media (which still can generate metabolites through mitochondria and PPP i.e. reviewed in Kalwat and Cobb 2017 Pharmacol Theraphy). I would consider these two point to slightly reformulate Lane 22 in the abstract, 176 and 187 in the discussion. I think might be more correct, considering the data presented in this work, to say that the insulin secretory pathway in DIO mice may be compromised by reduced oxidative pathway.
  • Lane 204 Please add the reference here on the previous studies you mention.
  • I wonder if data about similar comparison exist in the literature regarding human islets (i.e. after in vitro culture with glucose or palmitate) and if the authors can discuss the translation of this concept to human material? I wonder also if some work has been done by the authors in human islets since it is mentioned in the acknowledgments.

Author Response

We thank the reviewer for her/his valuable and constructive comments that we addressed point-by-point below.

Reviewer 2

In this work, Kabra et al. tested the hypothesis of correlation between respiratory rate and insulin secretion in pancreatic islets to evaluate dysfunctional aspect occurring during T2DM development. In my opinion, this work has important descriptive aspects that has an impact in the community studying metabolic alteration of pancreatic islets during T2DM and obesity development in relation of the ability to secrete insulin upon stimulation. The manuscript is enjoyable to read and well written. The introduction as well as the methods are clear and adequate to introduce the results. The hypothesis is well explained and is addressed very well in the discussion.

Very interesting and novel to my knowledge is the notion that after isolation of islets from DIO mice the GSIS is similar to control diet while the respiration in response to glucose stimulation is still affected. This work shows also that different ways to normalize the collected data might affect importantly the conclusion drawn by the same data. Together with the data showing an increase in insulin content (which might be due as the authors suggest by the increase in beta cells but also an increased insulin translation due to workload), the aspect studied here are often neglected but of high importance when studying the dysfunction of insulin secretion in DIO in mice. Moreover, the author clarify very well which aspect to consider (dimensionless data not needed to be normalized on cell mass or DNA which can increase the data spread) and which one can be predictive of dysfunction in these diet model.

I have few minor points that in my opinion need to be addressed before publication:

  • Please clarify which point you considered for the calculations of proton leak respiration and CE. Is it the maximal respiration after glucose stimulation (single highest point) and the lowest oligomycin point or is an average of several measurements?

RESPONSE: We thank this reviewer for the positive and expert evaluation. We state these important technical aspects now in the MS. Page 6 lines 194-203: ‘The individual bioenergetics parameters of OXPHOS were calculated as follows: Basal mitochondrial respiration = (last rate measured before glucose and /or other secretagogues injection) – (non-mitochondrial respiration rate). Mitochondrial respiration = (last rate measured before oligomycin injection) – (non-mitochondrial respiration rate). Proton leak-linked respiration = (minimum rate measured after oligomycin injection) - (non-mitochondrial respiration rate). ATP-linked respiration = mitochondrial respiration – proton leak-linked respiration. Coupling efficiency = represents the fraction of respiration used to drive ATP synthesis for each run, calculated as CE = 1- (proton leak-linked respiration / mitochondrial stimulated respiration).’.

  • Lane 28, 35, 53 β cells is spelled using β while in the rest of the text beta cells. I would suggest to be consistent in the nomenclature

RESPONSE: We thank for this notion and changed consistently to ‘β’ throughout the MS.

  • Lane 68. If the data were collected after the different diets were given for 16 weeks, can you clarify the reason of writing “for a maximum of 16w”.

RESPONSE: Indeed, all mice have been on the diets for 16 weeks and we have corrected this in the MS, page 6 line 165.

  • Lane 73. How long those islets were cultured before performing the Seahorse and insulin secretion experiments? This is a very critical point and need to be present in the method. In fact, in our experience after overnight culture the islets from DIO mice regain insulin secretion similar to control (when normalized on DNA) as the authors shows here, but retain some dysfunction if the insulin secretion is performed less than 4h after isolation. This point is important also to understand how long the islets from DIO mice retain dysfunctional aspect on their metabolism or if they can recover by longer exposure in vitro culture.

RESPONSE: We thank the reviewer for this important notion, and the fact that dysfunctional properties are seen shortly after incubation. We have performed the experiments after overnight incubation and included this information now on page 6, lines 173-175 ‘Islets were incubated overnight in RPMI 1640 culture medium supplemented with 10% (v/v) fetal calf serum (Life technologies) at 37°C and 5% CO2 before experimentation.’

  • Line 109 Please add if the insulin and glucose were measured in fasting or non-fasting.

RESPONSE:  We added this information on page 6 line 204: “Mice were fasted for six hours prior to blood collection”.

  • Lane 113 and 163 please correct normal (diet) with Chow.

RESPONSE: We have corrected this throughout the MS.

  • Could the author speculate why in DIO mice islets also the proton leak respiration is lower than controls?

RESPONSE: Yes, we speculate on compromised substrate oxidation, and added on page 3 lines 99-100 ‘The reduction of both proton leak and ATP-linked respiration can be indicative of compromised substrate oxidation capacity [8].’

  • Lane 140 The letter (I) can be placed at the start of the sentence “Correlation…”.

RESPONSE: This has been updated in the revised MS page 4 line 119.

  • Lane 145 the authors suggest that “the insulin values correlates significantly with ATP-linked respiration and GSR”. Wouldn´t be more suitable to say that the correlation between these parameter in chow diet and DIO mice are different? Along this line I would suggest, if appropriate for the authors, to add a dotted line with the regression of the DIO mice islets to show the difference compared to the chow regression in a visual format.

RESPONSE: Reviewer’s point well taken. We replaced the text on page 3 lines 104-105 with ‘This relationship appeared to be shifted upwards in DIO islets (dotted regression line).’ However, we finally did not add a dotted line for DIO islets, as the graph becomes too busy, given that we also state residual and correlation coefficient for the chow condition.

  • Line 175 The authors suggest that “the triggering pathway in DIO mice is compromised by reduced oxidative power, which could be mediated by impaired glucose uptake, glycolysis or pyruvate oxidation…”. I have some considerations here, which can be also included in the results or discussion. 1. The authors did not investigate the impairment of glucose uptake, glycolysis or pyruvate oxidation using an uncoupler (in separate experiment and not after glucose or oligomycin addition) which might have given an answer to this question thus avoiding the ATP/ADP regulation of substrate oxidation/utilization for cellular respiration. In fact, if the DIO mice are set to a higher ratio of cytosolic or mitochondria ATP/ADP ratio the substrate delivery and oxidation might be lower as shown here. 2. The insulin docked and secreted in the first phase is usually measured in the first 10-15min after glucose stimulation. Here the authors measure the insulin secretion after a static incubation of 1h and suggest that the triggering pathway is compromised. However this length of insulin secretion would include also the amplify pathway even though only glucose is present in the assay media (which still can generate metabolites through mitochondria and PPP i.e. reviewed in Kalwat and Cobb 2017 Pharmacol Theraphy). I would consider these two point to slightly reformulate Lane 22 in the abstract, 176 and 187 in the discussion. I think might be more correct, considering the data presented in this work, to say that the insulin secretory pathway in DIO mice may be compromised by reduced oxidative pathway.

RESPONSE: Reviewer’s point well taken and agreed. We changed the abstract to ‘Correlation of insulin secretion with respiratory parameters uncovers compromised insulin secretion in DIO islets by oxidative power.’ We changed the text of previous lines 176 and 187 to ‘secretory pathway’.

We added as lines 107-109 ‘However, we cannot formally exclude amplifying factors deriving from glucose catabolism given recent results by others [14], and we thus refer to altered secretory pathways of insulin secretion rather than to altered triggering of insulin secretion.’

  • Lane 204 Please add the reference here on the previous studies you mention

RESPONSE: We added the two references from our lab in the text (new line 231-232) and the reference list. 

  • I wonder if data about similar comparison exist in the literature regarding human islets (i.e. after in vitro culture with glucose or palmitate) and if the authors can discuss the translation of this concept to human material? I wonder also if some work has been done by the authors in human islets since it is mentioned in the acknowledgments.

RESPONSE: The half-sentence in the acknowledgement was a copy-paste error and has been removed – we have not performed respiration experiments for this study, nor published on respiration in human islets previously. In previous studies, we have performed insulin secretion experiments on human islets, but material was too limited for respiration measurements.  Preliminary, unpublished data of a related collaboration (characterizing beta cell differentiation states), however, indicated that defective insulin secretion is related to CE, but these data of post-mortem samples were highly variable. With new Seahorse gadgets, e.g. spheroid perfusion plates, there is a real chance to reduce amount of material and consolidate the relation between respiration and insulin secretion in human islets. We have not systematically screened the literature for examples of human islets vs respiration.